REGISTERED REPORT PROTOCOL

# Peer support workers in co-production and co-creation in public mental health and addiction services: Protocol for a scoping review

**Kristina Bakke Aakerblom** [1]*, **Ottar Ness** [2]

**1** Department of Welfare and Participation, Faculty of Health and Social Sciences, Western Norway University of Applied Research, Bergen, Norway, **2** Department of Education and Lifelong Learning, Norwegian University of Science and Technology, Trondheim, Norway

* krbaa@hvl.no

## Abstract

Worldwide, there is a growing interest to employ people with lived experiences in health and social services. Particularly in mental health and addiction services, individuals with lived experience of mental health problems enter the workplace as peer support workers (PSW´s). Their aim in the services is to bring in the perspective of service users in interactive processes at the micro and macro levels. The services´ ability to exploit the knowledge from PSW´s lived experiences will influence both the content and quality of the services, its effectiveness and its capacity to innovate and change. The concepts of co-production and co-creation are used to describe these interactive processes in the services in the literature. While co-production is aimed at improving individual services, co-creation seeks to develop service systems. This scoping review aims to provide an overview of the research status of PSW´s different involvement, in co-production and co-creation, in public mental health and addiction services. Studies describing PSW´s involvement in co-production and co-creation will be contrasted and compared. Knowledge about PSW´s involvement in co-production and co-creation is vital for understanding and further developing these interactive processes with PSW´s. The studies reviewed will describe PSW´s different types of involvement in co-production and co-creation in public mental health and addiction services or across organizational and institutional boundaries. The research question is: How are peer support workers involved in co-production and co-creation in public mental health and addiction services, and what are the described outcomes? Literature searches are conducted in Medline, PsycINFO, Embase, Oria, WorldCat, Google Scholar, Scopus, Academic Search Elite, Cinahl, and Web of Science, from the inception of each database to January 4, 2021. Expected results are that PSW´s are often described as a frontline worker who spends most of their working hours in a joint effort to co-produce with service users. Fewer studies describe PSW´s involved in interactive processes to re-design or transform public services systems. It is anticipated that this scoping review will increase the knowledge of the services' abilities to exploit PSW´s expertise and inform policy and research.

**Data Availability Statement:** All relevant data from this study will be made available upon study completion.

**Funding:** KBAA has a doctoral scholarship from the Dam foundation. Dam Foundation have 38 members, all voluntary health and rehabilitation organizations based in Norway. The foundation receives part of the profit from the national lottery in Norway, Norsk Tipping. The funds from the lottery supports project work in voluntary humanitarian organizations in preventive health care, rehabilitation and research. All voluntary humanitarian/socially dedicated organizations and professional organizations for the functionally disabled in Norway may apply for funds regardless of membership in the foundation.

**Competing interests:** The authors have declared that no competing interests exist.

# Introduction

In public mental health and addiction services, an increasing number of individuals with lived experience of mental health problems enter the workplace as peer support workers (PSW´s). There is a growing range of research on PSW´s [1]. Existing research seems to be primarily focused on the normative side and demonstrates why we should implement PSW´s. The research literature indicates that deployment of PSW´s leads to increased service user participation [2], active involvement and empowers service users. This makes the services more democratic and lead to realizing mental health as a human right [1].

The introduction of PSW´s in the public mental health and addiction services seems to fit like a hand in a glove with a service-dominant approach to public sector innovation [3]. At the service production level, the service users are understood as the nodal point of discretionary service production. The key is to unlock the "tacit knowledge" that service users possess to re-design the entire service system in the pursuit of producing public value. Understanding service users´ perspectives, and translating it into policies and practice, is perceived as a valuable source of increasing public mental health systems´ responsiveness to service users´ needs and goals. For this purpose, one particular strategy is integrating PSW´s in the mental health workforce [4].

In the research literature, co-production and co-creation are concepts used to describe such interactive processes. Sometimes these concepts are used interchangeably without distinguishing between their meanings [5]. This scoping review will make a conceptual distinction between them based on a service-dominant approach [3] to public sector innovation. Co-production then refers to the interactive process through which the service providers and service users apply their different resources and capabilities in service production and delivery. The purpose of the joint interactive process in co-production is to produce and deliver a pre-determined public service. While the service may be adjusted and improved to meet the users´ needs, it is not subject to innovation defined as the development and realization of new disruptive ideas [6]. A service-dominant approach to public sector innovation takes the process beyond individual services co-production and unlocks the potential for the co-design and co-creation of public services. Co-creation often involve more participants (volunteers, social entrepreneurs, citizens, private service actors) than the dyadic interplay of service producers and service users in co-production. Co-creation is considered a vital tool in enhancing innovation and improving the relevance of services. Co-production and co-creation are both gaining ground at the public service production level. However, most public services are still designed and produced by bureaucratic and professionalized public agencies with little contribution from service users, citizens, and volunteers [7].

A scoping review is chosen because it is suited to get an overview of PSW´s different involvement in public mental health and addiction services. Scoping reviews can be useful when bringing together literature in disciplines with emerging evidence. Furthermore, it is suited to address questions beyond effectiveness and efficiency [8]. PSW´s involvement in interactive processes is often described using other terms than co-production or co-creation. Therefore, it is necessary to have a broad search in the research literature. The research will be mapped with a particular objective to obtain an overview of the types of co-production and co-creation PSW´s are involved in, the conditioning factors (antecedents and barriers), and described aims and actual outcomes.

## Aims of the scoping review

In the research literature, PSW´s often appear to engage directly in co-production in service delivery with service-users. To be able to add value to the production process in the pursuit of

public value outcomes, service-users need to be involved in the early stages of public service development, i.e., in the initiation, decision-making and design phases [9]. Following this, we assume that the different types of interactive processes PSW´s are involved in can influence their ability to contribute to innovative changes and at the other side; the services´ ability to utilize PSW´s knowledge in developing and realizing new disruptive ideas. Co-creation processes have recently been found to have an innovative dimension not found in co-production [10]. A systematic review of citizens involvement in co-production and co-creation distinguishes between three different types, "an initiator, a co-implementor or a co-designer" which differ in their degree of involvement. We assume that PSW´s different involvement in co-production and co-creation will influence the content and quality of services for the service users, as well as the effectiveness and bottom line of organizations and services' ability to innovate and change. The relevance of this scoping review will be twofold. First, we will explore what types of involvement PSW´s take on, typically described with different "co"; co-initiated, co-design, co-deliver, co-implement or co-evaluate. Secondly, we will compare and contrast their different involvement with its influential factors (antecedents and barriers), described aim and reported outcomes. This will be critical knowledge to be able to make use of PSW´s competence and develop the interactive processes with PSW´s in the services.

## Methods

The data in this review will be collected based on Arkzey and O'Malley´s framework [11], with four phases: 1) identify the research questions, 2) search for relevant studies, 3) select studies, 4) chart the data, and 5) collate, summarize and report studies. This scoping review´s research question is: How are peer support workers involved in co-production and co-creation in public mental health and addiction services and, what are the described outcomes? Last literature search will be 2021.01.04. The protocol was registered in Protokols.io: 2021.02.11. The study protocol and, all available data, will be made public when the study is conducted and published.

### Eligibility criteria

**Types of studies.**   Articles should describe interactive processes with PSW´s in either the design, development, delivery, implementation or evaluation of services in public mental health and addiction services. Articles that do not describe interactive processes, but describe its outcomes, will be included if it is clear that PSW´s were involved.

The public sector will be defined broadly as those parts of the economy that are either in-state ownership or services under contracts to the state, because countries differ in what degree mental health services are part of the state or operate with contracts to the state. We will include different designs and aims of services such as acute services, outreach services, community services, social welfare services, hospital services, activity-based services, public management services, etc.

**Focus of studies.**   All studies describing types of interactive processes and/or PSW´s different positions, roles, tasks, or activities in the services and across organizational or institutional boundaries, will be included. Articles describing mutual peer support, self-help groups, consumer-driven services, peer counselling, peer-led education or peer counselling programs related to medical or physical conditions, will not be included.

**Participants.**   Participants engaged in interactive processes with PSW´s, like co-production or co-creation, in public mental health and addiction services. In a service setting a) members of the public who might be using a service or intervention; service-users, careers, patients, relatives, b) people working in the services; PSW´s, professional staff and managers, c) citizens

or their representatives, social entrepreneurs, volunteers, d) organizations (voluntary, private and public) and/or organizations representing service users. When participants are people, we will include only adults from the general population (age 18–65).

**Study design.** Studies will not be restricted to a particular study type.

**Language.** Only articles written in English will be included.

**Publication status.** Only international peer-reviewed journal articles and books will be included.

## Study selection

When reporting, the following tool will be used: the "Preferred Reporting Items for Systematic reviews and Meta-Analyses extension for Scoping Reviews (PRISMA-ScR) Checklist" [8]. A flow diagram will be used to detail the study selection process (Fig 1).

**Search strategies for the identification of studies.** Database sources for the review are Medline, PsycINFO, Embase, Oria, WorldCat, Google Scholar, Scopus, Academic Search Elite, Cinahl, and Web of Science. The search is limited to title, abstract, and keywords. The titles or abstracts for all studies included in the database searches will be read. In addition to searching in electronic databases, we will search manually within reference lists and perform citation searches of the included studies and authors to identify further publications linked to the included articles. We will also consult experts in peer support work in mental health and addiction services. The search in databases will be from the inception of each database chosen until 04.01.2021.

## The search will, in all the databases, consist of the following search terms

1. peer group

2. peer adj (provid* or support*)).ti,ab.

3. (live* adj experience*).ti,ab.

4. psw.ti,ab.

5. (expert adj by adj experienc*).ti,ab.

6. prosum*.ti,ab.

7. enduce*.ti,ab.

8. (boundary adj spanner*).ti,ab.

9. (peer adj mentor*).ti,ab.

10. (peer adj educator*).ti,ab.

11. (peer adj advocate*).ti,ab.

12. (peer adj listen*).ti,ab.

13. (peer adj provid*).ti,ab.

14. or/1-13

15. exp Cooperative Behavior/

16. collaborat*.ti,ab.

17. participat*.ti,ab.

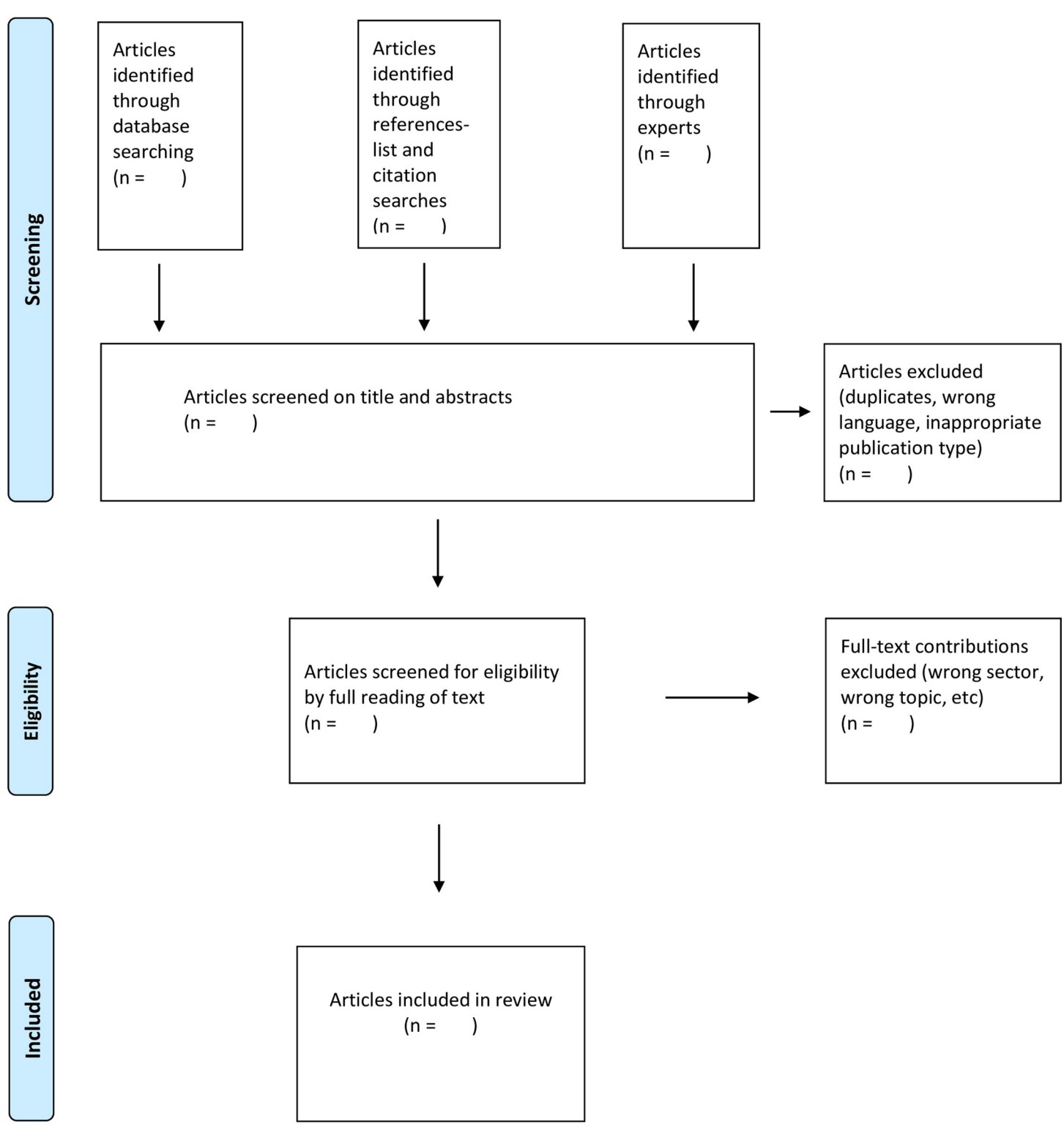

**Fig 1. PRISMA flow diagram of the search strategy.**

18. integrat*.ti,ab.

19. ((collaborat* or social) adj innovat*).ti,ab.

20. cooperat*.ti,ab.

21. cocreat*.ti,ab.

22. (co adj creat*).ti,ab.

23. coproduct*.ti,ab.

24. (co adj produc*).ti,ab.

25. or/15-24

26. exp Public Sector/

27. exp Health Care Sector/

28. exp Mental Health Services/

29. exp Mental Health/

30. exp State Medicine/

31. exp Primary Health Care/

32. exp "Delivery of Health Care"/

33. (public adj care adj service*).ti,ab.

34. (public adj service*).ti,ab.

35. (mental adj health*).ti,ab.

36. (Addiction adj Service*).ti,ab.

37. exp Health Services/

38. (Peer adj Recovery adj Support adj Service*).ti,ab.

39. (recover* adj service*).ti,ab.

40. municipal*.ti,ab.

41. (Social adj health adj care*).ti,ab.

42. exp Social Work/

43. (Social adj service*).ti,ab.

44. (statutory adj mental adj health adj service*).ti,ab.

45. exp Community Mental Health Services/

46. (third adj sector adj organisation*).ti,ab.

47. or/26-46

48. 14 and 25 and 47

49. limit 48 to English

**Data synthesis.** The mapping will follow a funnel approach, meaning that the studies will be categorized in stages. For each step, the number of studies in each category will be counted

and described. The number of included studies will thus decline through the mapping process until only the relevant studies remain, which will be read and mapped in detail.

**Data extraction.** The articles will be handled using Endnote and the online software for conducting literature reviews, Rayyan [12]. First, all records from the databases will be downloaded to Endnote. From Endnote, the records will be exported to Rayyan. Rayyan will be further used as a working tool for the review. The first author (KBAA) will perform the first steps in the selection process, which includes (1) removing duplicates, (2) removing studies that are not written in English, (3) removing editorials, reports, opinion papers, and conference papers, (4) removing studies that are not conducted in or related to mental health services. All these steps will be decided based on reading the abstract. If it is not possible to determine based on this, the studies will be included. The remaining records will then be read by the first author (KBAA), and the second author (ON) will read 20% of all the records randomly. Rates of concordance between these researchers will be calculated, and any disagreements will be discussed and resolved together. The focus of discussion will be if the articles clearly describe interactive processes and whether PSW´s involvement in the process is clearly explained. If needed, disagreements will further be discussed and solved with the PhD candidates extended supervisory team, as this review article is part of a PhD project. The next step will be to read all included articles to identify if they meet the eligibility criteria. This step will be performed by the first author (KBAA). The second author (ON) will read 20% of all the studies randomly. Both review authors will then read all the included articles based on a full reading of articles and decide which to be included in the review. As this is a scoping review, our goal is to determine the available range of studies (quantitative and/or qualitative). We also intend to address the current state of knowledge, offer a new perspective and point out areas for future research, in line with a "state of the art"-review [13]. We will represent this visually as a mapping or a charting of the located data. However, if data is suitable for quantitative analysis at this stage, a meta-analysis will be considered.

## Information on the following categories will be sought in the studies

1. First author-name

2. Year of publication

3. Country

4. Process of collaboration (co-production and/or co-creation)

5. Participants (PSW´s, service-users, careers, professionals, leaders, volunteers, services, organizations)

6. PSW´s various types of involvement (co-produce, co-deliver, co-implement, co-initiated, co-design, co-evaluate)

7. PSW´s positions (administrative functions, caseworker, frontline worker)

8. Aims of PSW´s involvement

9. Influential factors (antecedents and barriers)

10. Outcome (for; service-users, PSW´s, services, organizations)

11. Context

12. Study design (methodology and data analysis)

13. Study findings/results

14. Comments

## Discussion

In this study protocol, we make a conceptual distinction between co-production and co-creation. While co-production is aimed at improving individual services, co-creation seeks to develop service systems. A reason to separate the concepts is to illuminate that co-creation processes have an innovative dimension not found in co-production [10]. A service-dominant approach to public sector innovation highlights the need to utilize PSW´s tacit knowledge to re-design service systems and pursue public value outcomes. In that case, we should pay attention to their involvement in co-creation versus co-production processes.

Research indicates that PSW´s most often are involved in the direct delivery of services. This co-producer role has proven to be essential because it contributes to greater involvement from service users [2] in addition to more satisfied service users. Less research seems to describe PSW´s involvement in co-creation processes, the re-design or development of service systems. Based on a service-dominant approach to public sector innovation [3], we argue that PSW´s different types of involvement will influence their ability to contribute to innovation and change in the services. A recent study demonstrates that contributions from staff with addiction experience in SUD treatment to collaborative and responsive service production had bounded efficacy and limited influence over organization-level changes [14]. To influence the services delivery systems, PSW´s need to be involved in the initiation, decision-making and design phases [9].

Participation in decision-making is still often felt beyond certain people's capabilities [15]. The role of service-users and professional service providers shows very clear contrasts in different public administration regimes, especially in determining service quality [16]. How the interactive processes with PSW´s develop may depend on the interplay of forces at the micro and macro levels of society favoring the development of particular public administration regimes rather than another. Internationally, there is a great diversity in the implementation of PSW´s, and the tendency worldwide is nevertheless towards employed of PSW´s in a diverse and growing number of domains, such as education, politics, social sectors, healthcare and welfare, and research [17]. Many countries are in the process of engaging PSW´s in their mental health and addiction services. Therefore, it is increasingly relevant to study the actual involvement of PSW´s within the interactive processes. This has the potential for developing in quite different directions in different public administrative regimes. It is reasonable to expect that the interactive processes will develop both in an individual or collective fashion and that it will involve more or less citizens participation, depending on the public administration regime [16]. This implies that citizens and PSW´s involvement in co-production and co-creation processes will be both regime specific and service specific.

### Limitation of the study

We have chosen to do a scoping review because we assume that it will not be possible to make a statistical comparison of PSW´s different involvement in the public mental health and addiction services. The reviews will not assess the studies´ quality, and the reliability of data extracted from selected studies will not be commented on. PSW´s as an employer group is relatively new in the services, and too renders true for the research field. Much of the existing research on PSW´s seem to focus on the normative side primarily and demonstrate why we should implement PSW´s. Seemingly, less attention is paid to PSW´s involvement or roles in

the interactive processes and the outcomes of PSW´s different contributions. This is a limitation. In this review, we have chosen to use the term "peer support worker" due to this title being common across countries. People with lived experience of mental health and/or addiction experiences working in mental health and addiction services hold a wide range of different job titles. This implies that we may possibly fail to grasp all the literature describing this employer group, which is another limitation. In this review, we apply a framework from public sector innovation studies and elaborate on PSW´s involvement in interactive processes described in the research literature. The literature describing PSW´s contributions often depict the interactive processes with other terms than co-production and co-creation. Altogether, the authors are required to read all the research articles that describe PSW´s different involvement, and then decide, based on the various descriptions and concepts used, which interactive processes PSW´s are involved in. We make a conceptual distinction between the concepts of co-production and co-creation, while the existing research literature seems to use the concepts interchangeably. This may also be a limitation of the study.

## Quality assessment /risk of bias and evidence quality

A scoping review is designed to provide an overview of the existing research literature. We will not perform a formal assessment of the methodological quality of the included studies. The exception is if meta-analysis is conducted at the final stage (see paragraph above); then the risk of bias and strength of evidence will be assessed for the studies included in the meta-analysis. The included papers´ quality will then be rated according to an adapted version of the Critical Appraisal Skills Programme (CASP) for qualitative research [18].

## Acknowledgments

This protocol for a scoping review is related to the first authors´ PhD research project. The project addresses peer support workers' integration in public mental health and addiction services as a co-creative social innovation.

## Author Contributions

**Conceptualization:** Kristina Bakke Aakerblom.

**Formal analysis:** Kristina Bakke Aakerblom, Ottar Ness.

**Investigation:** Kristina Bakke Aakerblom.

**Methodology:** Kristina Bakke Aakerblom.

**Project administration:** Kristina Bakke Aakerblom.

**Resources:** Kristina Bakke Aakerblom.

**Supervision:** Ottar Ness.

**Validation:** Kristina Bakke Aakerblom, Ottar Ness.

**Visualization:** Kristina Bakke Aakerblom.

**Writing – original draft:** Kristina Bakke Aakerblom, Ottar Ness.

**Writing – review & editing:** Kristina Bakke Aakerblom, Ottar Ness.

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
