## [Decision Letter · Decision Letter 0]

6 Jan 2021

PONE-D-20-37362

The influence of peer support worker integration on co-production and co-creation in public mental health and addiction services: Protocol for a scoping review

PLOS ONE

Dear Kristina Bakke Aakerblom,

Thank you for submitting your manuscript to PLOS ONE. After careful consideration, we feel that it has merit but does not fully meet PLOS ONE’s publication criteria as it currently stands. Therefore, we invite you to submit a revised version of the manuscript that addresses the points raised during the review process.

We look forward to receiving your revised manuscript.

Kind regards,

Marianne Storm

Academic Editor

PLOS ONE

Journal Requirements:

Reviewers' comments:

Reviewer's Responses to Questions

**Comments to the Author**

1. Does the manuscript provide a valid rationale for the proposed study, with clearly identified and justified research questions?

Reviewer #1: Yes

Reviewer #2: Partly

2. Is the protocol technically sound and planned in a manner that will lead to a meaningful outcome and allow testing the stated hypotheses?

Reviewer #1: Yes

Reviewer #2: Yes

3. Is the methodology feasible and described in sufficient detail to allow the work to be replicable?

Reviewer #1: Yes

Reviewer #2: Yes

4. Have the authors described where all data underlying the findings will be made available when the study is complete?

Reviewer #1: No

Reviewer #2: Yes

5. Is the manuscript presented in an intelligible fashion and written in standard English?

Reviewer #1: Yes

Reviewer #2: Yes

6. Review Comments to the Author

You may also provide optional suggestions and comments to authors that they might find helpful in planning their study.

Reviewer #1: Review for minor grammar and spelling errors.

Reason for "major review" selection: Include a discussion at the end for the potential impact of the scoping review on the field. This is important to include.

Remove info on how data will be collected in the “aim of study” section. This more appropriately belongs in “methods.”

Include the visual you discussed about the study selection process, inclusion, and exclusion.

PSW or PSWs? In some sentences, plural seems to fit in better with certain sentences and grammar.

Spell out all acronyms the first time you use them (United Nations, line #65).

Need to describe where data will be made available when the study is complete.

Include clear dates of articles included in study. Currently says from inception to 8.31.2020, but that you will update again 1.4.2021. Does that mean that your search will include items that are published up until January 2021? Or will the search only go up until the end of August 2020? Also define what inception is. Is it the start of the journal, etc.?

Reviewer #2: Thank you for the opportunity to review this Registered Report Protocol that aims to examine the influence of peer support worker integration on co-production and co-creation in public health and addictions services. This is an interesting study with some important research questions. It is very important to understand how peer support workers are integrated into co-production and co-creation processes in health system. My comments are below regarding this manuscript:

1. Abstract is not traditional written, which is fine. I see authors report background and method. Final sentence in abstract is research question. I believe abstract will be improved if authors add a one or two sentence/s regarding any expected results and and how these results will inform field.

2. Although authors defined co-creation and co-production, It is still difficult to understand the difference between co-creation and co-production. The distinction between co-creation and co-production could be explained with an example.

3. Second paragraph in introduction has some parts that could be moved to purpose of study. For example, line 93 and rest is about purpose of study. Interestingly, in "Aim of the Study" section authors talk about background again. I suggest authors to move all background information into introduction section. I also suggest authors have a brief and concise purpose of study.

4. There are too many questions in introduction. It is very hard to overall understand introduction due to all questions and how introduction is organized. I suggest authors to be more concise with purpose section and improve background section.

5. I wonder whether this following sentence should move to Method: "The data will be collected based on Arkzey and O'Malley's framework (10): 1) identify the research questions, 2) search for relevant studies, 3) select studies, 4) chart the data, and 5) collate, summarize and report studies."

6. What year period authors aim to conduct this study? Any time restriction such as from 2010 to 2020.

7. In any disagreement, what is the plan? I see authors reported that they will have discussion on disagreement to resolve it. What will be the focus of discussion? What is further step if there is no resolution? For example, do authors plan to bring a third reviewer to help resolution.

8. I see that editorials, reports, and opinion papers will be removed. Later, in "Information on the Following Categories....", item 4 has opinion papers and others.

Thank you for letting me review this Registered Report Protocol. This study seems to be an interesting and informing study.

7. PLOS authors have the option to publish the peer review history of their article (what does this mean?). If published, this will include your full peer review and any attached files.

Reviewer #1: No

Reviewer #2: No

---

## [Author Response · Author response to Decision Letter 0]

17 Feb 2021

Rebuttal Letter: Response to Reviewers PLOS ONE

Manuscript title: 

Peer support workers in co-production and co-creation in public mental health and addiction services: Protocol for a scoping review

Dear PLOS ONE Academic Editor and Reviewers,

Thank you for your very useful comments which provide very valuable insights to improve our current manuscript quality and coherency. Each of the statements given is well analyzed and provided with necessary explanation, correction and response to support the manuscript's points. Hence, below we addressed your comments and responses. First, we will address the reviewer´s response to Questions 1 and 4, then we will address each reviewer´s specific comments:

Responses to question 1 and 4:

1. Does the manuscript provide a rationale for the proposed study, with a clearly identified and justified research questions? 

The research question outlined is expected to address a valid academic problem or topic and contribute to the base of the knowledge in the field. 

Reviewer 2# Partly. 

We have revised the research question to: How are peer support workers involved in co-production and co-creation in public mental health and addiction services and, what are the described outcomes? 

This has also led to a refinement of the title to “Peer support workers in co-production and co-creation in public mental health and addiction services: Protocol for a scoping review”

4. Have the authors described where all data underlying the findings will be made available when the study is complete? 

Reviewer 1: No. 

Author comments: This information has been updated and answered in the comments no 6, below. The protocol was registered in Protokols.io: 2021.02.11. The study protocol and, all available data, will be made public when the study are conducted and published. 

The study protocol and, all available data, will be made public when the study finishes. 

Comments and responses to each reviewer’s feedback:

Reviewer 1:

Thank you for your thorough feedback. Your comments made us able to deliver an enhanced and better manuscript. To respond clearly to all comments, they are numbered so that it is easier to follow: 

1) Reason for "major review" selection: Include a discussion at the end for the potential impact of the scoping review on the field. 

Authors response: A discussion at the end of the research protocol is included. The discussion part is characterized by the fact that it is a research protocol. This implies that we have pointed to the reasons for doing this study and what we expect to find based on our research question and why this knowledge will be of importance for policy, practice and research. 

2) Remove info on how data will be collected in "the aim of study" section. This more appropriately belongs in "methods". 

Authors response: Information on how the data will be collected are removed from "Aims of scoping review" to "Method". 

3) Include the visual you discussed about the study selection process, inclusion and exclusion. 

Response: A visualization of the study selection process based on PRISMA-Scr is added.

4) PSW or PSWs? In some sentences, plural seems to fit better with certain sentences and grammar. 

Authors response: We have revised and consequently use PSW's. This fits better with grammar, as you suggest. However, the plural form works best because we address PSW's as employees in the services and not the individual employee. 

5) All acronyms are spelled out the first time when they are used. For instance, (United Nations, line #). 

Authors response: Thank you for making us aware of this mistake. Due to the refinement of this section, this sentence is no longer part of the introduction. 

6) Need to describe where data will be made available when study is complete.

Authors response: The method section is updated with this information: The protocol was registered in Protokols.io: 2021.02.11. The study protocol and, all available data, will be made public when the study are conducted and published. 

The study protocol and, all available data, will be made public when the study finishes. 

7) Include clear dates of articles included in study. Also define what inception is. Is it the start of journal, etc.? 

Authors response: This information is updated in the Method section, under the subheading, “study selection”: The search in databases will be from the inception of each database chosen until 04.01.2021. 

Comments & Response for Reviewer 2:

Thank you for your thorough and useful feedback. Your comments made us able to revise and enhance the quality and arguments in our manuscript. Here, we will address your feedback: 

Comment 1: Abstract is not traditional written, which is fine. I see authors report background and method. Final sentence in abstract is research question. I believe abstract will be improved if authors add a one or two sentences regarding any expected results and how these results will inform field.

Authors response: We have tried to improve the abstract, regarding sentences on expected results and how these results will inform field are included at the end. The text has been refined, and the abstract is more traditionally written. 

Comment 2: Although authors defined co-creation and co-production. It is still difficult to understand the difference between co-creation and co-production. The distinction between co-creation and co-production could be explained with an example.

Authors response: The difference between the two concepts is now more precise defined in the text. First, in the abstract and then, this has been explained more thoroughly in the introduction, with examples. 

We will address comment 3 and 4 together:

Comment 3: Second paragraph in introduction has some parts that could be moved to purpose of study. For example, line 93 and rest is about purpose of study. Interestingly, in "Aim of the Study" section authors talk about background again. I suggest authors to move all background information into introduction section. I also suggest authors have a brief and concise purpose of study. 

Comment 4: There are too many questions in introduction. It is very hard to overall understand introduction due to all questions and how introduction is organized. I suggest authors to be more concise with purpose section and improve background section.

Authors responses to comments 3 and 4: The introduction section has been substantially modified. There are no questions in the introduction. The introduction starts with a background section where the research status and theoretical and practical assumptions for this study are accounted for. The reason to split between the concept, co-production and co-creation is explained, as the rationale for doing this study and in choosing this approach. The last section in the introduction describes the aim of this scoping review. We argue that PSW's different involvement in the interactive processes can influence their ability to contribute to innovative changes and that we need to pay attention to their involvement in co-creation processes because they have an innovative dimension not found in co-production. 

Comment 5: I wonder whether this following sentence should move to Method: "The data will be collected based on Arkzey and O'Malley's framework (10): 1) identify the research questions, 2) search for relevant studies, 3) select studies, 4) chart the data, and 5) collate, summarize and report studies." 

Authors response: This sentence has been moved to the Method section. 

Comment 6: What year period authors aim to conduct this study? Any time restriction such as from 2010 to 2020. 

Authors response: This is corrected in the Method section to be: From the inception of each included database until January 04.01.2021. 

Comment 7: In any disagreement, what is the plan? I see the authors reported that they would have a discussion on disagreement to resolve it. What will be the focus of discussion? What is further step if there is no resolution? For example, do authors plan to bring a third reviewer to help resolution. 

Authors response: This is corrected to be: The focus of discussion will be if articles clearly describe interactive processes, and if PSW's involvement in the process is clearly explained. If needed, disagreements will further be discussed and solved with the PhD candidates extended supervisory team, as this review article is part of a PhD project. 

Comment 8: I see that editorials, reports, and opinion papers will be removed. Later, in "Information on the Following Categories....", item 4 has opinion papers and others. 

Authors response: Thank you for making us aware of this mistake. We will not keep information about opinion papers. This has been deleted from the sentence.

---

## [Editor Report · Decision Letter 1]

2 Mar 2021

Peer support workers in co-production and co-creation in public mental health and addiction services: Protocol for a scoping review

PONE-D-20-37362R1

Dear Dr. Aakerblom,

We’re pleased to inform you that your manuscript has been judged scientifically suitable for publication and will be formally accepted for publication once it meets all outstanding technical requirements.

Kind regards,

Sherief Ghozy, M.D., Ph.D. candidate

Academic Editor

PLOS ONE

Additional Editor Comments:

There are many errors of English grammar and use in this manuscript, I would highly recommend cop-editing of this manuscript prior to final disposition.

---

## [Editor Report · Acceptance letter]

5 Mar 2021

PONE-D-20-37362R1 

Peer support workers in co-production and co-creation in public mental health and addiction services: Protocol for a scoping review 

Dear Dr. Aakerblom:

I'm pleased to inform you that your manuscript has been deemed suitable for publication in PLOS ONE. Congratulations! Your manuscript is now with our production department. 

Kind regards, 

on behalf of

Dr. Sherief Ghozy 

Academic Editor

PLOS ONE